# Prevalence of pregnancy termination and associated factors among married women in Papua New Guinea: A nationally representative cross-sectional survey

**McKenzie Maviso**[1]*, **Paula Zebedee Aines**[2], **Gracelyn Potjepat**[2], **Nancy Geregl**[3], **Glen Mola**[4], **John W. Bolnga**[5]

1 Division of Public Health, School of Medicine and Health Sciences, University of Papua New Guinea, Port Moresby, Papua New Guinea, 2 Division of Nursing, School of Medicine and Health Sciences, University of Papua New Guinea, Port Moresby, Papua New Guinea, 3 School of Health Sciences, Pacific Adventist University, Port Moresby, Papua New Guinea, 4 Department of Obstetrics and Gynaecology, School of Medicine and Health Sciences, University of Papua New Guinea, Port Moresby, Papua New Guinea, 5 Department of Obstetrics and Gynaecology, Modilon Hospital, Madang, Papua New Guinea

* mckenzie.maviso@upng.ac.pg

**Data Availability Statement:** Permission for accessing and analyzing the 2016–2018 PNGDHS data was obtained from the DHS Program. All data and DHS-related materials are available from the

## Abstract

### Background

Pregnancy termination or induced abortion is not decriminalized, and access to safe abortion services is largely unavailable in Papua New Guinea (PNG). However, the practice is common throughout the country. This study aimed to estimate the prevalence and determine factors associated with pregnancy termination among married women aged 15–49 years in PNG.

### Methods

Secondary data from the 2016–2018 PNG Demographic and Health Survey (PNGDHS) was used. A total weighted sample of 6,288 married women were included. The Complex Sample Analysis method was used to account for the cluster design and sample weight of the study. Chi-square tests and multivariable logistic regression were used to assess factors associated with pregnancy termination. Adjusted odds ratios (aORs) with 95% Confidence Intervals (CIs) were reported.

### Results

The prevalence of pregnancy termination was 5.3%. Nearly half (45.2%) of all pregnancy terminations occurred in the Highlands region. Women aged 35–44 years (aOR = 8.54; 95% CI: 1.61–45.26), not working (aOR = 6.17; 95% CI: 2.26–16.85), owned a mobile phone (aOR = 3.77; 95% CI: 1.60–8.84), and lived in urban areas (aOR = 5.66; 95% CI: 1.91–16.81) were more likely to terminate a pregnancy. Women who experienced intimate partner violence (IPV) were 2.27 times (aOR = 2.27; 95% CI: 1.17–4.41) more likely to terminate a pregnancy compared to those who did not experience IPV. Women with unplanned

DHS team upon request at https://dhsprogram.com/.

**Funding:** The author(s) received no specific funding for this work.

**Competing interests:** The authors have declared that no competing interests exist.

pregnancies were 6.23 times (aOR = 6.23; 95% CI: 2.61–14.87) more likely to terminate a pregnancy. Women who knew about modern contraceptive methods and made independent decisions for contraceptive use were 3.38 and 2.54 times (aOR = 3.38; 95% CI: 1.39–8.18 and aOR = 2.54; 95% CI: 1.18–5.45, respectively) more likely to terminate a pregnancy.

### Conclusion

The findings highlight the role of sociodemographic and maternal factors in pregnancy termination among married women in PNG. Efforts aimed at reducing unplanned pregnancies and terminations should focus on comprehensive sexual and reproductive health education and improving easy access to contraceptives for married couples. Post-abortion care should also be integrated into the country's legal framework and added as an important component of existing sexual and reproductive health services.

### Introduction

While every pregnancy sets a woman at risk of death, the vulnerability to disability and maternal mortality is greater among women whose pregnancies are terminated through induced abortions, miscarriages, or stillbirths than among those who have live births [1, 2]. Pregnancy termination, also known as induced abortion, is a medical or surgical intervention that involves removing a viable fetus, whereas spontaneous abortions (or miscarriages) occur when an embryo or fetus is lost due to natural causes [3]. In addition, induced abortion is permitted when there are compelling reasons, such as to save a woman's life, prevent adverse physical and mental health outcomes, avoid pregnancy following rape or incest, prevent serious fetal anomalies, socioeconomic reasons, or upon a woman's request [4–6]. Conversely, unsafe abortion is the termination of pregnancy performed by unskilled persons in an environment lacking primary medical and standard sanitary conditions, or both [7].

Recent global data showed that between 2015 and 2019, an estimated 121 million pregnancies that occurred were unintended [8]. Of these unintended pregnancies, 61% (73.3 million) resulted in abortion, consistent with a global abortion rate of 39 per 1,000 women aged 15–49 years [8]. An estimated 45% of all abortions are unsafe, and 97% occur in low- and middle-income countries (LMICs) [4, 8]. Likewise, about 4.7–13.2% of all global maternal deaths each year are attributed to unsafe abortion [4, 9]. Disability and deaths associated with unsafe pregnancy termination persist as a public health burden, particularly in settings where health disparities are evident, and/or abortion laws are restrictive [8, 10, 11]. Similarly, in countries where patriarchal societies, cultural norms, religious beliefs, and economic factors influence women's decisions, pregnancy termination is never easy [12–14]. The practice remains prevalent in low- and middle-income countries (LMICs), where women of reproductive age (15–49 years) have high rates of unmet contraceptive needs [8, 15]. On average, women in these countries have more pregnancies throughout their lifetime, and their risk of pregnancy-related disability and mortality remains higher than those living in high-income countries [8]. Unintended pregnancy can have substantial social, economic, psychological, and health consequences for women of reproductive age and their families [16–18].

Many countries still confront health challenges due to the complexity of abortion laws and poor abortion healthcare services, and even where abortions are legal, the policies are hindered

by the practice or incoherence [11, 19]. Unsafe abortion has adverse health and economic implications for individuals and society [9, 20]. Studies have revealed that unintended pregnancy has substantial social, economic, psychological, and health consequences for women of reproductive age and their families [16–18]. An earlier World Health Organization (WHO) systematic analysis highlighted five primary complications that accounted for 75% of all pregnancy-related deaths, which include hemorrhage, infection, pre-eclampsia and eclampsia, birth complications, and unsafe abortion [9]. It can also be aggravated by specific predisposing co-morbidities such as obesity, asthma, diabetes, and hypertension in women with a higher risk of pregnancy termination [21–24].

Despite the high risk of disability and death, the choice to undergo a pregnancy termination is personal and may be affected by several circumstance-specific considerations and healthcare services. For instance, evidence from population-based surveys in LMICs revealed that demographic (e.g., age, marital status, parity, education), socioeconomic (e.g., employment), health risk behaviors (e.g., substance abuse, sexual activities), and sociocultural (e.g., gender norms, belief systems) factors are associated with pregnancy termination [10, 14, 25]. Previous experience with pregnancy termination, access to reproductive health services, and attitudes towards this practice have also been well documented [13, 25]. Along with adverse health and socioeconomic consequences, decision-making regarding pregnancy termination is problematic, notably where patriarchy, restricted abortion laws, cultural and religious beliefs, and economic factors may impact women's decisions, especially in resource-constrained settings with already overloaded health systems [11].

In PNG, the risk of maternal and neonatal deaths due to pregnancy- and childbirth-related complications is disproportionately higher compared to other countries in the Western Pacific region [26, 27]. The maternal mortality ratio is estimated to be 215–545 deaths per 100,000 live births [26, 28]. Sepsis due to unsafe abortion is one of the leading causes of maternal mortality after hemorrhage in the country [27, 29]. Regarding contraceptive use, the country has a low prevalence rate for modern contraceptive methods among married women (31%), and a high unmet need for family planning (26%) [30].

While pregnancy termination or induced abortion for socioeconomic reasons or upon request is prohibited under PNG's *Criminal Code Act 1974* [31], it is performed by a professional medical practitioner under certain conditions; for example, if the pregnancy is caused by rape or incest, if continuing the pregnancy places a woman's life or her physical or mental health at significant risk, or if the child may experience a severe physical abnormality or disease [31]. Nonetheless, this practice continues to be contested in the country, with vested interests from politically and religiously conservative spheres, patriarchal societies, and sociocultural belief systems [31, 32]. Moreover, the majority of women are unaware of the legal implications following pregnancy termination, especially illegal and unsafe abortion, which is known to be widely performed in the country [33–35]. Evidence from PNG sources suggests that many women of reproductive age (15–49 years) with unplanned pregnancies perform unsafe abortions, which may result in abortion-related morbidity and mortality [26, 29, 33, 36]. Previous reviews in the Eastern Highlands Province found that unsafe abortion infection accounted for 48% of maternal mortality [29], and 24% of women who sought post-abortion care were due to unsafe abortions [35]. Another study in Madang found that more than three-quarters (76.5%) of abortion-related admissions were due to continuous bleeding following an induced abortion [37]. Pregnancy-related complications, including unsafe abortion practices, contribute to the country's poor maternal health indicators.

Considering the morbidity and mortality associated with the complications of pregnancy terminations, particularly unsafe abortions in PNG, it is imperative to fully understand the prevalence and associated factors to adequately address this issue. Although a few studies have

determined the factors associated with pregnancy termination, country-level estimates of the prevalence and their determinants are largely undetermined, affecting a significant proportion of women in the reproductive age group [14]. Therefore, this study aimed to estimate the prevalence and determine factors associated with pregnancy termination among married women aged 15–49 in PNG. This could encourage program planning and policy development to establish a well-defined legal framework for implementing abortion services in the country.

## Materials and methods

### Study setting

This study used secondary data from the PNG Demographic and Health Survey (PNGDHS), a nationally representative population-based survey conducted across the four major administrative regions (Southern, Highlands, Momase, and Islands) from October 2016 to December 2018.

### Study design and data source

Data were drawn from the PNGDHS 2016–2018, conducted every five years, employing a stratified two-stage cluster sampling design. In the first stage, census blocks are selected with systematic proportional and stratified sampling by urban and rural areas (except for the National Capital District, which does not have rural areas). In the second stage, a fixed number of 24 households per cluster are selected with an equal probability of systematic selection. A total of 17,505 households were selected for the sample, of which 16,754 were occupied. Of the occupied households, 16,021 were interviewed. In the interviewed households, 18,175 women were identified for interviews, and 15,198 women participated successfully, yielding a response rate of 84%. The data used in the analyses were weighted to explain variations in the probability of selection and non-response. In this study, a weighted sub-sample consisted of 6,288 married women aged 15–49 years who were married or in a formal union and who had ever been pregnant, with complete cases on all of the variables studied included. Women with missing information, who had never been married and never had sex were excluded from the study because they had no risk of terminating a pregnancy (Fig 1). Detailed sampling procedures have been reported [30].

### Definition of variables

**Dependent variable.** The dependent variable for this study was pregnancy termination. Women were asked whether they had ever terminated a pregnancy. Information about the dependent variable was generated from this question. Respondents provided a "yes" or "no" as a response to the question to indicate whether they had ever terminated a pregnancy or not. Based on their responses, a dichotomous response of 'Yes' was coded '1' when a woman reported that she had terminated a pregnancy, and 'No' was coded '0' if she had not.

**Independent variables.** The independent variables were selected based on their availability in the dataset, practical significance, and theoretical relevance reported in the literature about pregnancy termination [14, 38, 39]. Variables were categorized into maternal, household, and maternal health-related characteristics. Maternal characteristics include age, educational level, literacy, occupation, mobile phone ownership, internet use, and place of residence. Household characteristics include the husband's age, educational level, and occupation, the household wealth index, and the number of children living. Additionally, participants were asked whether they had experienced intimate partner violence (IPV) during their pregnancies. Their responses were categorized as having experienced IPV due to one or more of the

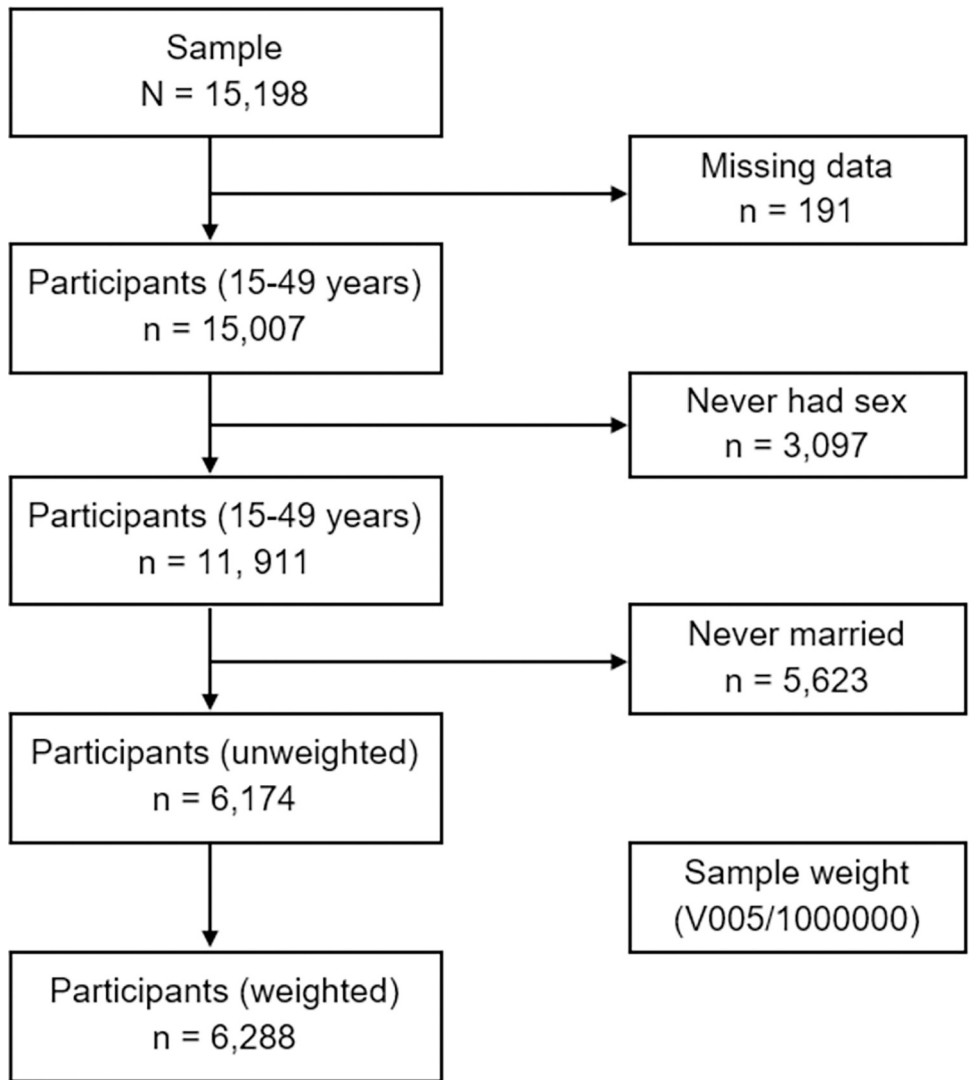

**Fig 1. Flowchart of participant selection for this study.**

following: being humiliated, threatened, insulted, pushed, slapped, punched with a fist, or hit with an object; being kicked, burned, threatened, or attacked with a knife; being twisted; being physically forced to have sexual intercourse; being physically forced to have any other sexual act; or performing any sexual acts against their will during pregnancy [30]. Maternal health-related characteristics include the last pregnancy being planned, knowledge of modern contraceptive methods, decision-maker for contraceptive use, knowledge of the menstrual cycle, number of antenatal visits, births in the last three years, and birth order.

## Operational definition

Pregnancy termination or induced abortion: a pregnancy that is terminated by choice through an intervention. For this study, the terms pregnancy termination and abortion are used interchangeably.

## Statistical analysis

Analyses were restricted to participants with complete data for the variables of interest. Sample weights were used in all analyses to adjust for disproportionate sampling and obtain reliable estimates and standard errors. Descriptive statistics were computed to report frequencies and percentages of the sample's characteristics. The Chi-square test of independence was used to assess the association between the dependent and independent variables. Variables with a p<0.2 in the bivariate analysis were retained and included in the multivariate logistic regression analysis. The variance inflation factor was used to check for multicollinearity, and there was no evidence of multicollinearity. The model's fitness was checked using Hosmer and Lemeshow [40]. To account for the multistage sampling design and sample weight, the Complex Samples Analysis technique was used, which provided generalizable and accurate estimates of proportion, probability values, and odds ratios [41]. Adjusted odds ratios (aORs) with 95% confidence intervals (CIs) were reported. A p ≤0.05 was considered statistically significant. Analyses were performed using IBM SPSS Statistics for Windows, Version 26.0 (IBM Corp., Armonk, NY, USA).

## Ethical considerations

Permission to access the datasets was obtained from the DHS Program and was only used for this study and not shared with a third party. The Institutionalized Review Board (IRB), the DHS survey implementing agency, and the IRB in host countries approved the survey protocols, ensuring ethical conduct in accordance with human subject research. Further ethical approval for this study was not required since data are available in the public domain (https://dhsprogram.com/). Written informed consent was obtained from the participants before each interview [30]. All the information about the participants had been anonymized before accessing the dataset for final analysis.

# Results

## Characteristics of the study participants

A total of 6,288 married women were included in the study. The mean age was 29.13 (± 9.61) years. Nearly half of the women were aged 25–34 years (47.9%) and had primary education (49.1%). The majority of them lived in rural areas (80.6%) and were not working (66.8%). Over half (51.9%) reported that they had experienced IPV from their husbands. Most women who planned their last pregnancies (80.6%) knew about modern contraceptive methods (86.4) and had menstrual cycle knowledge (77.7%). Similarly, over half (59.6%) of them made a joint decision (with their husbands) about contraceptive use and had one pregnancy (63.3%) in the past three years preceding the survey (Table 1).

## Prevalence of pregnancy termination

The prevalence of pregnancy termination was 5.3% (95% CI: 0.05–0.06) among married women of reproductive age (15–49 years) in PNG. Among the three administrative regions, the Highlands (45,2%) had the highest pregnancy terminations rates (Fig 2).

## Bivariate analysis of factors of pregnancy termination

Among women who had terminated a pregnancy, half (50.7%) of them were between the ages of 25 and 34 years. Many (42.6%) had primary education, while 53% were unemployed. Similarly, 43% of women's husbands had primary education, and 64.3% were employed. The majority (71.5%) experienced intimate partner violence. Less than half (46.2%) of the

**Table 1. Characteristics of the study participants (N = 6,288).**

| Characteristics | Frequency (n) | Percent (%) |
|---|---|---|
| *Maternal factors* | | |
| Age (years) | | |
| 15–24 | 1,536 | 24.4 |
| 25–34 | 3,010 | 47.9 |
| 35–44 | 1,561 | 24.8 |
| 45–49 | 181 | 2.9 |
| Mean age (SD) = 29.13 (± 9.61) | | |
| Educational level | | |
| No formal education | 1,612 | 25.6 |
| Primary | 3,082 | 49.1 |
| Secondary | 1,336 | 21.2 |
| Higher | 258 | 4.1 |
| Literacy (n = 6,259) | | |
| Cannot read/write | 2,312 | 36.9 |
| Can read/write | 3,947 | 63.1 |
| Occupation | | |
| Not working | 4,199 | 66.8 |
| Working | 2,089 | 33.2 |
| Tobacco/cigarette smoking | | |
| No | 4,920 | 78.2 |
| Yes | 1,368 | 21.8 |
| Mobile phone ownership | | |
| No | 4,362 | 69.4 |
| Yes | 1,926 | 30.6 |
| Place of residence | | |
| Urban | 703 | 11.2 |
| Rural | 5,585 | 88.8 |
| *Household factors* | | |
| Age of husband (years) (n = 3,916) | | |
| 15–24 | 312 | 8.0 |
| 25–34 | 1,371 | 35.0 |
| 35–44 | 1,308 | 33.4 |
| 45 or more | 925 | 23.6 |
| Educational level (husband) (n = 4,251) | | |
| No formal education | 920 | 21.6 |
| Primary | 1,929 | 45.4 |
| Secondary | 1,165 | 27.4 |
| Higher | 237 | 5.6 |
| Occupation (husband) (n = 5,582) | | |
| Not working | 2,773 | 49.7 |
| Working | 2,809 | 50.3 |
| Wealth index | | |
| Poorest | 1,335 | 21.2 |
| Poorer | 1,246 | 19.8 |
| Middle | 1,245 | 19.8 |
| Richer | 1,250 | 19.9 |
| Richest | 1,212 | 19.3 |

*(Continued)*

**Table 1.** (Continued)

| Characteristics | Frequency (n) | Percent (%) |
|---|---|---|
| Number of children living | | |
| 1 | 1,461 | 23.2 |
| 2 | 1,294 | 20.6 |
| 3 or more | 3,533 | 56.2 |
| Intimate partner violence (IPV) (n = 2,500) | | |
| No | 1,203 | 48.1 |
| Yes | 1,297 | 51.9 |
| *Maternal health-related factors* | | |
| Last pregnancy planned | | |
| No | 1,216 | 19.3 |
| Yes | 5,072 | 80.7 |
| Knowledge of modern contraceptive methods | | |
| No | 855 | 13.6 |
| Yes | 5,433 | 86.4 |
| Knowledge of the ovulatory cycle | | |
| No | 1,401 | 22.3 |
| Yes | 4,887 | 77.7 |
| Decision-maker for contraceptive use (n = 1,646) | | |
| Respondent | 454 | 27.6 |
| Husband/partner | 199 | 12.0 |
| Joint decision | 993 | 60.4 |
| Number of antenatal visits (n = 2,655) | | |
| No visits | 642 | 24.2 |
| 1–3 | 563 | 21.2 |
| 4 or more | 1,450 | 54.6 |
| Birth order | | |
| 1st | 1,356 | 21.6 |
| 2nd | 1,246 | 19.8 |
| 3rd | 1,178 | 18.7 |
| 4th or more | 2,508 | 39.9 |
| Births in the last 3 years (n = 4,661) | | |
| 1 | 3,978 | 85.4 |
| 2 | 658 | 14.1 |
| 3 | 25 | 0.5 |

respondents made independent decisions regarding contraceptive use. The Chi-square analysis revealed that maternal age, educational level, occupation, mobile phone ownership, internet use, place of residence, husband's occupation, household wealth index, number of children living, IPV experience, knowledge of modern contraceptive methods, decision-maker for contraceptive use, ovulatory cycle knowledge, birth order, and births in the last three years were found to be statistically significant factors of pregnancy termination ($p < 0.05$) (Table 2).

## Multivariable analysis of factors associated with pregnancy termination

In the multivariable analysis, maternal age, education, occupation, mobile phone ownership, place of residence, IPV, knowledge of modern contraceptive methods, decision-maker for contraceptive use, and number of children living were significantly associated with pregnancy termination.

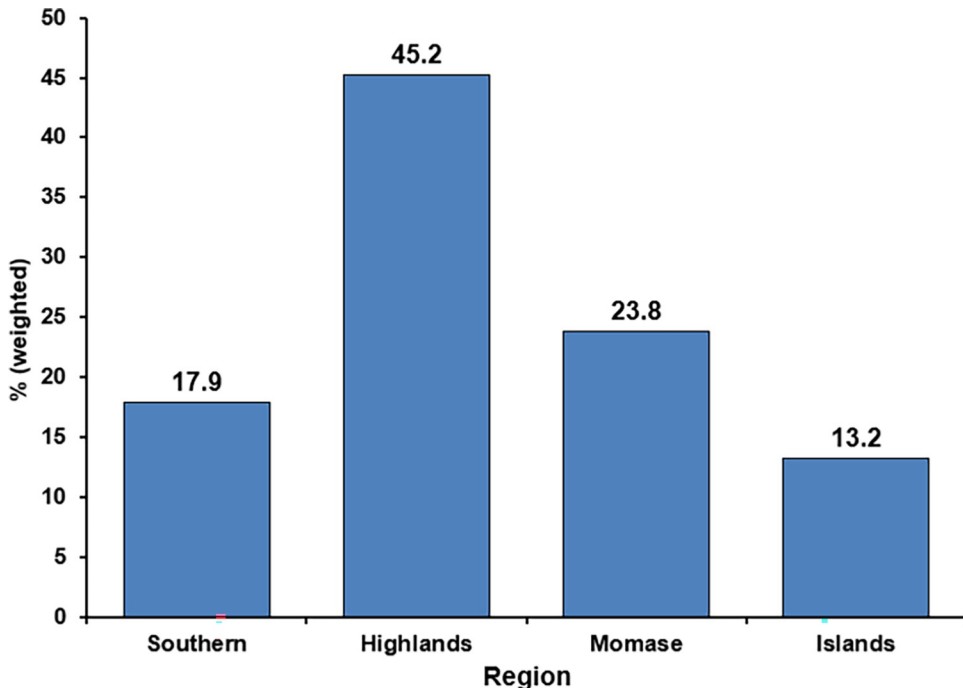

**Fig 2. The rates of pregnancy terminations by region in PNG.**

Women aged 35–44 years (aOR = 8.54; 95% CI: 1.61–45.26), not working (aOR = 6.17; 95% CI: 2.26–16.85), owned a mobile phone (aOR = 3.77; 95% CI: 1.60–8.84), and lived in urban areas (aOR = 5.66; 95% CI: 1.91–16.81) were more likely of pregnancy termination. Women who experienced IPV were 2.27 times (aOR = 2.27; 95% CI: 1.17–4.41) more likely to terminate a pregnancy compared to those who did not experience IPV. Women with unplanned pregnancies were 6.23 times (aOR = 6.23; 95% CI: 2.61–14.87) more likely to terminate a pregnancy. Women who knew about modern contraceptive methods and made independent decisions for contraceptive use were 3.38 and 2.54 times (aOR = 3.38; 95% CI: 1.39–8.18 and aOR = 2.54; 95% CI: 1.18–5.45, respectively) more likely to terminate a pregnancy. However, those aged 15–24 years (aOR = 0.62; 95% CI: 0.07, 5.26), had husbands not working (aOR = 0.27; 95% CI: 0.12, 0.63), had husbands making decisions for contraceptive use (aOR = 1.33; 95% CI: 0.54, 3.28), and had two children living (aOR = 0.69; 95% CI: 0.96, 5.02) were less likely to terminate a pregnancy (Table 3).

## Discussion

The current study used nationally representative survey data, and estimated the prevalence and determined factors of pregnancy terminations among married women of reproductive age in PNG. It was found that married women endure a disproportionate proportion of pregnancy terminations that remain prevalent in the country, despite their social and legal implications [34]. In multivariable logistic regression analysis, maternal age, maternal occupation, mobile phone ownership, place of residence, husband's occupation, IPV, pregnancy being planned, knowledge about modern contraceptive methods, and decision-maker for contraceptive use were found to be statistically significant factors in pregnancy termination.

The overall prevalence of pregnancy termination was 5.3%, with higher proportions reported from PNG's Highlands region (45.2%). The prevalence of pregnancy termination was

**Table 2. Bivariate analysis of factors of pregnancy termination (N = 6,288).**

| Characteristics | Pregnancy termination | | | | |
|---|---|---|---|---|---|
| | **No** | **%** | **Yes** | **%** | **p-value*** |
| **Sample** | **5,948** | **94.7** | **340** | **5.3** | |
| *Maternal factors* | | | | | |
| Age (years) | | | | | **0.023** |
| 15–24 | 1,473 | 24.8 | 63 | 18.6 | |
| 25–34 | 2,837 | 47.7 | 172 | 50.7 | |
| 35–44 | 1,463 | 24.6 | 98 | 28.9 | |
| 45–49 | 175 | 2.9 | 6 | 1.8 | |
| Educational level | | | | | **<0.001** |
| No formal education | 1,558 | 26.2 | 54 | 15.9 | |
| Primary | 2,937 | 49.4 | 145 | 42.6 | |
| Secondary | 1,246 | 20.9 | 90 | 26.5 | |
| Higher | 207 | 3.5 | 51 | 15.0 | |
| Literacy (n = 6,259) | | | | | **<0.001** |
| Cannot read/write | 2,220 | 37.5 | 92 | 27.1 | |
| Can read/write | 3,700 | 62.5 | 247 | 72.9 | |
| Occupation | | | | | **<0.001** |
| Not working | 4,019 | 67.6 | 180 | 52.9 | |
| Working | 1,929 | 32.4 | 160 | 47.1 | |
| Tobacco/cigarette smoking | | | | | **0.001** |
| No | 4,680 | 78.7 | 240 | 70.8 | |
| Yes | 1,268 | 21.3 | 99 | 29.2 | |
| Mobile phone ownership | | | | | **<0.001** |
| No | 4,159 | 69.9 | 203 | 59.7 | |
| Yes | 1,789 | 30.1 | 137 | 40.3 | |
| Place of residence | | | | | **<0.001** |
| Urban | 633 | 10.3 | 70 | 20.6 | |
| Rural | 5,315 | 89.4 | 270 | 79.4 | |
| *Household factors* | | | | | |
| Age of husband (years) (n = 3,917) | | | | | 0.624 |
| 15–24 | 296 | 8.0 | 17 | 8.7 | |
| 25–34 | 1,299 | 34.9 | 72 | 36.7 | |
| 35–44 | 1,251 | 33.6 | 57 | 29.1 | |
| 45 or more | 875 | 23.5 | 50 | 25.5 | |
| Educational level of husband (n = 4,251) | | | | | 0.320 |
| No formal education | 880 | 21.8 | 40 | 18.7 | |
| Primary | 1,837 | 45.5 | 92 | 43.0 | |
| Secondary | 1,095 | 27.1 | 70 | 32.7 | |
| Higher | 225 | 5.6 | 12 | 5.6 | |
| Occupation of husband (n = 5,582) | | | | | **<0.001** |
| Not working | 2,672 | 50.4 | 101 | 35.7 | |
| Working | 2,627 | 49.6 | 182 | 64.3 | |
| Wealth index | | | | | **<0.001** |
| Poorest | 1,278 | 21.5 | 58 | 17.1 | |
| Poorer | 1,204 | 20.2 | 42 | 12.4 | |
| Rich | 1,176 | 19.8 | 70 | 20.6 | |
| Richer | 1,177 | 19.8 | 72 | 21.2 | |

(*Continued*)

**Table 2.** (Continued)

| Characteristics | Pregnancy termination | | | | |
|---|---|---|---|---|---|
| | **No** | **%** | **Yes** | **%** | **p-value*** |
| Richest | 1,113 | 18.7 | 98 | 28.8 | |
| Number of children living | | | | | **<0.001** |
| 1 | 1,418 | 23.8 | 43 | 12.7 | |
| 2 | 1,219 | 20.5 | 74 | 21.8 | |
| 3 or more | 3,312 | 55.7 | 222 | 65.5 | |
| Experienced intimate partner violence (IPV) (n = 2,500) | | | | | **<0.001** |
| No | 1,158 | 49.4 | 45 | 28.5 | |
| Yes | 1,184 | 50.6 | 113 | 71.5 | |
| *Maternal health-related factors* | | | | | |
| Last pregnancy planned | | | | | 0.306 |
| No | 1,143 | 19.2 | 73 | 21.5 | |
| Yes | 4,805 | 80.8 | 267 | 78.5 | |
| Knowledge of modern contraceptive methods | | | | | **<0.001** |
| No | 840 | 14.1 | 15 | 4.4 | |
| Yes | 5,108 | 85.9 | 325 | 95.6 | |
| Knowledge of the ovulatory cycle | | | | | **0.025** |
| No | 1,342 | 22.6 | 59 | 17.4 | |
| Yes | 4,606 | 77.4 | 281 | 82.6 | |
| Decision-maker for contraceptive use (n = 1,646) | | | | | **<0.001** |
| Respondent | 405 | 26.2 | 48 | 47.1 | |
| Husband/partner | 180 | 11.7 | 19 | 18.6 | |
| Joint decision | 959 | 62.1 | 35 | 34.3 | |
| Number of antenatal visits (n = 2,655) | | | | | 0.660 |
| No visits | 618 | 24.3 | 24 | 21.4 | |
| 1–3 | 536 | 21.1 | 27 | 24.1 | |
| 4 or more | 1,389 | 54.9 | 61 | 54.5 | |
| Birth order | | | | | **<0.001** |
| 1st | 1,320 | 22.2 | 36 | 10.6 | |
| 2nd | 1,185 | 19.9 | 61 | 17.9 | |
| 3rd | 1,078 | 18.1 | 100 | 29.4 | |
| 4th or more | 2,365 | 39.8 | 143 | 42.1 | |
| Births in the last 3 years (n = 4,661) | | | | | **0.002** |
| 1 | 3,778 | 85.7 | 200 | 78.7 | |
| 2 | 608 | 13.8 | 50 | 19.7 | |
| 3 | 21 | 0.5 | 4 | 1.6 | |

*Chi-square test, p ≤0.05

relatively lower than that of studies from East Africa (7.8%) [39], including Ethiopia (8.5%) [10], and Sierra Leone [42]. This difference may have existed due to the study population, geography, and the accessibility and availability of sexual and reproductive health services, including maternal and child health programs over the years.

According to the study's results, being older was one of the major factors associated with pregnancy termination, particularly among women aged 35 or more. This finding was consistent with the results of prior studies conducted in 36 LMICs [14], including Ghana [43], Mozambique [43], and Sierra Leone [42], where women of advanced age had increased odds

**Table 3. Multivariable analysis of factors associated with pregnancy termination (N = 6,288).**

| Characteristics | cOR (95% CI) | aOR (95% CI) | p-value* |
|---|---|---|---|
| Age (years) | | | <**0.001** |
| 15–24 | 1.34 (0.47, 3.81) | 0.62 (0.07, 5.26) | |
| 25–34 | 1.89 (0.84, 4.28) | 1.00 (0.17, 5.95) | |
| 35–44 | 2.09 (0.94, 4.66) | 8.54 (1.61, 45.26) | |
| 45–49 | Ref. | Ref. | |
| Educational level | | | 0.292 |
| No formal education | Ref. | Ref. | |
| Primary | 1.42 (0.86, 2.34) | 0.89 (0.30, 2.65) | |
| Secondary | 2.07 (1.23, 3.48) | 1.12 (0.21, 5.81) | |
| Higher | 7.10 (2.85, 17.68) | 0.30 (0.05, 2.00) | |
| Literacy | | | 0.523 |
| Cannot read/write | Ref. | Ref. | |
| Can read/write | 1.62 (1.11, 2.35) | 1.44 (0.47, 4.39) | |
| Occupation | | | <**0.001** |
| Not working | 0.54 (0.37, 0.79) | 6.17 (2.26, 16.85) | |
| Working | Ref. | Ref. | |
| Tobacco/cigarette smoking | | | 0.487 |
| No | Ref. | Ref. | |
| Yes | 1.53 (0.89, 2.63) | 1.38 (0.55, 3.45) | |
| Mobile phone ownership | | | **0.002** |
| No | Ref. | Ref. | |
| Yes | 1.57 (1.06, 2.33) | 3.77 (1.60, 8.84) | |
| Place of residence | | | **0.002** |
| Urban | 2.19 (1.53, 3.13) | 5.66 (1.91, 16.81) | |
| Rural | Ref. | Ref. | |
| Occupation (husband) | | | **0.003** |
| Not working | 0.54 (0.37, 0.81) | 0.27 (0.12, 0.63) | |
| Working | Ref. | Ref. | |
| Wealth index | | | 0.113 |
| Poorest | Ref. | Ref. | |
| Poorer | 0.77 (0.41, 1.44) | 1.17 (0.43, 3.20) | |
| Rich | 1.31 (0.78, 2.19) | 0.44 (0.13, 1.44) | |
| Richer | 1.35 (0.79, 2.32) | 0.19 (0.04, 0.88) | |
| Richest | 1.95 (1.06, 3.59) | 0.19 (0.04, 0.98) | |
| Number of children living | | | **0.033** |
| 1 | Ref. | Ref. | |
| 2 | 1.99 (1.19, 3.32) | 0.69 (0.96, 5.02) | |
| 3 or more | 2.20 (1.36, 3.57) | 0.07 (0.01, 0.95) | |
| Experienced intimate partner violence (IPV) | | | **0.016** |
| No | Ref. | Ref. | |
| Yes | 2.48 (1.52, 4.04) | 2.27 (1.17, 4.41) | |
| Last pregnancy planned | | | <**0.001** |
| No | 1.14 (0.77, 1.69) | 6.23 (2.61, 14.87) | |
| Yes | Ref. | Ref. | |
| Knowledge of modern contraceptive methods | | | **0.007** |
| No | Ref. | Ref. | |
| Yes | 3.65 (1.81, 7.36) | 3.38 (1.39, 8.18) | |

(*Continued*)

**Table 3.** (Continued)

| Characteristics | cOR (95% CI) | aOR (95% CI) | p-value* |
|---|---|---|---|
| Knowledge of the ovulatory cycle | | | 0.923 |
| No | Ref. | Ref. | |
| Yes | 1.39 (0.88, 2.20) | 0.96 (0.41, 2.27) | |
| Decision-maker for contraceptive use | | | **0.05** |
| Respondent | 3.29 (1.19, 9.09) | 2.54 (1.18, 5.45) | |
| Husband/partner | 2.89 (1.26, 6.65) | 1.33 (0.54, 3.28) | |
| Joint decision | Ref. | Ref. | |
| Birth order | | | 0.394 |
| 1st | 0.45 (0.27, 0.74) | 0.07 (0.03, 1.68) | |
| 2nd | 0.85 (0.59, 1.22) | 0.39 (0.06, 2.54) | |
| 3rd | 1.54 (0.82, 2.88) | 0.59 (0.23, 1.53) | |
| 4th or more | Ref. | Ref. | |
| Birth in the last 3 years | | | 0.443 |
| 1 | Ref. | Ref. | |
| 2 | 1.57 (0.63, 3.88) | 0.48 (0.16, 1.48) | |
| 3 | 3.23 (0.66, 15.73) | – | |

*p $\leq$ 0.05

cOR = crude Odds Ratio; aOR = adjusted Odds Ratio; CI = Confidence Interval; Ref. = Reference category.

of pregnancy terminations compared to young women. A possible explanation could be that advanced maternal age predisposes women to medical and pregnancy-related complications such as pre-eclampsia, ectopic pregnancy, and gestational diabetes, including the unavoidable nature of maternal age, which may complicate pregnancy and lead to ending a pregnancy [14, 44, 45]. The positive association between married women and increased pregnancy termination rates could be due to the ineffectiveness or lack of contraceptive use [46]. Similarly, women who have reached their desired family size and believe they cannot get pregnant at that age have higher odds of terminating a pregnancy. A perceived need or a lack of access to contraceptive use at the end of the reproductive years could be the possible reason [42].

The association between place of residence and pregnancy termination has received much consideration in numerous studies, demonstrating that women residing in urban areas are significantly more likely to terminate a pregnancy [10, 14, 37]. Compared to rural women in this study, those in urban areas were about six times more likely to terminate a pregnancy. Urban women may have access to abortion information and services, including self-induced abortion [47]. On the contrary, the lower likelihood of pregnancy termination among rural women may be elucidated by the fact that access to abortion services, including sexual and reproductive health care, is inadequate because of disparities in health services and resource allocations between rural and urban areas. Similarly, restrictive gender and sociocultural norms that influence the sexual and reproductive healthcare-seeking behavior of rural women may be a possible explanation for the low prevalence of pregnancy termination.

The current study found a statistically significant relationship between pregnancy termination and occupation, with the odds higher among women who were not working. These findings are comparable to previous studies [43, 48], indicating a higher concentration of pregnancy terminations among socioeconomically disadvantaged women. Women who reported pregnancy termination may do so for financial constraints, partner-related reasons, or a desire to postpone childbearing, as demonstrated by prior studies [49, 50]. Moreover,

financial barriers may contribute to disparities in contraceptive usage among poor women, who may not be able to afford modern contraceptives compared to working women, resulting in terminating a pregnancy. In contrast, other studies found that working women had higher probabilities of terminating a pregnancy than their unemployed counterparts [25, 42, 43]. Studies have argued that educated and working women are financially empowered, prioritize employment continuity, have a greater awareness of contraceptive options, and can afford abortion services [42, 43].

Compared to women who do not own mobile phones, the likelihood of terminating a pregnancy remained high among those who own mobile phones. This finding is contrary to previous studies [51–53], indicating that mobile phones in health (mHealth) have been shown to influence women's perceptions and decisions on pregnancy and abortion services and improve post-abortion care, including family planning. Mobile phone use has increased considerably, along with reducing its costs, and remains a popular means to seek healthcare or obtain information about health issues [54]. In addition, women who have access to social media may be aware of the abortion laws in their country and are less likely to be stigmatized by society in their quest to have a pregnancy terminated [43]. Further research is needed to determine the relationship between the use of mobile phones and pregnancy termination among married women.

The association between IPV and pregnancy termination revealed in this study bolsters the earlier findings that being in an abusive relationship may affect women's reproductive decision-making, which can result in pregnancy termination [55–57]. Married women who experienced IPV were twice as likely to terminate a pregnancy, which corroborates a recent study from PNG [58]. It has been reported that over two-thirds of PNG women have suffered some form of physical or sexual violence in their lifetime [59]. Women in abusive relationships may likely have less control over their sexuality and, as a result, become pregnant more often than they should, which might lead to an increase in the incidence of pregnancy terminations [56, 57]. Another possible explanation could be that the husband may be unwilling to accept the child and may use violence or other coercive means to force the woman to terminate the pregnancy or negatively influence her decision. Moreover, women experiencing IPV may have less autonomy in sexual and reproductive health and are confronting the challenges of unmet contraceptive use and unintended pregnancies [55, 58, 60]. While terminating a pregnancy may most likely be the woman's choice, other options for her may also be limited in an abusive relationship. An insight into socioeconomic and demographic conditions sanctioning violence remains crucial and entails women's empowerment programs. The call to reinforce the *National Strategy to Prevent and Respond to Gender-Based Violence* [61], and prioritize violence prevention, along with strengthening provisions for issuing interim and long-term protection orders for survivors is necessary. Furthermore, empowering women through education and social support services can enhance their self-confidence and enable them to make informed decisions about sexual and reproductive health.

In this study, unplanned pregnancy was a significant predictor associated with increased odds of pregnancy termination. This is consistent with a previous study conducted in Kenya [62]. One elucidation may be that unintended pregnancy mainly results from a lack of or inconsistent use of contraceptive methods, as reported in earlier studies [63, 64]. Similarly, women may have insufficient or inaccurate knowledge and concerns about the side effects of contraception. Evidence has shown that contraception use is influenced by women's knowledge, beliefs, perceptions of health risks, and previous experience [65]. The inadequate provision of family planning services to address contraceptive needs for married women at risk of unintended pregnancies may lead to increased unwanted pregnancies, followed by pregnancy termination [65, 66]. Effective contraception has several advantages, including better mother

health and social and economic empowerment, while reducing the risk of an unplanned or undesired pregnancy [65]. In addition, the need to improve family planning services while paying special attention, particularly for women of advanced age or those who prefer not to have children is warranted.

The current study complements the growing evidence that women who knew about modern contraceptive methods were three times more likely to terminate a pregnancy compared to their counterparts. This association corroborated recent findings in Ethiopia [10] and Nepal [67], where women with excellent contraceptive and abortion knowledge were more likely to terminate a pregnancy than those with poor contraceptive knowledge. Similarly, this study found a significant association between individual decision-making regarding contraceptive use and pregnancy termination. Women who could make independent decisions about their reproductive health were more likely to terminate a pregnancy [25]. Women may foresee financial constraints associated with childrearing, social pressure against untimely pregnancies, and experiences of marital issues that encourage them to terminate a pregnancy [58, 68]. Another possible explanation could be that women who are educated and working have higher odds of decision-making power [69].

Concerning birth history, women with two or fewer children were less likely to terminate a pregnancy. This suggests that women understand the importance of contraceptive use and have delayed childbearing or achieved the desired family size. Consistent with studies in sub-Saharan Africa [14, 70], pregnancy termination rates remained low among women with fewer than four children. The probable reason may be that women with fewer children could do so due to effective contraceptive use, higher education attainment, and the probability of their involvement in income-generating activities. Another possible explanation is that women with living children may have a decreased future reproductive desire and a high intention to use contraceptives [71].

## Strengths and limitations of the study

The study had several strengths. Data were drawn from a nationally representative survey conducted from 2016 to 2018, which was weighted to ensure representativeness and a valid estimate. The study also employed the Complex aSmples Analysis method to account for the multistage sampling design used in the PNGDHS and obtained a reliable standard error and estimate. However, this survey was cross-sectional, and the results cannot be used to make causal inferences. There is also a high probability of social desirability bias, which might result in underreporting. A plausible explanation is that women may be reluctant to admit to terminating a pregnancy since it is not decriminalized and for fear of experiencing intimate partner violence from their partners, where social and cultural norms influence women's decisions [34, 58]. Furthermore, the PNGDHS database did not include methods of pregnancy termination with abortifacients such as misoprostol or mifepristone or any other non-conventional methods such as traditional herbs, barks, leaves, or traumatic physical injuries. Nonetheless, the questions were generalized retrospectively to establish the prevalence of pregnancy termination only.

## Implications for policy and practice

Women who terminate a pregnancy outside the purview of the *Criminal Code Act* [31] are considered offenders in PNG. Abortion has not been decriminalized in the country, and the law only outlines the circumstances under which it will not be considered illegal. Nonetheless, the growing evidence of abortion-related complications requiring hospitalization, including disability and mortality, suggests that the practice remains prevalent and may increase. In

consultation with qualified medical practitioners and professional bodies (the Professional Medical and Nursing/Midwifery Society and the National Department of Health), they need to recommend constitutional directives and reformation of the criminal law restricting access to termination of pregnancy. Furthermore, decriminalizing abortion should be viewed within a medical-legal paradigm while ensuring that the service is accessible and available to reduce adverse maternal health outcomes in the country.

## Conclusion

The findings highlight the role of sociodemographic and maternal factors in pregnancy termination among married women in PNG. Maternal age, occupation, mobile phone ownership, living in urban areas, IPV, unplanned pregnancy, knowledge of modern contraceptive methods, and decision-maker for contraceptive use were significantly associated with pregnancy termination. Abortion-related complications and, in some cases, deaths are due to the criminalization of this health care issue, and it is inconsistent with human rights that this should be more so for the disadvantaged, the uneducated, the lower socioeconomic strata of society, and women living in rural areas. Efforts aimed at reducing unplanned pregnancies and terminations should focus on improving easy access to contraceptives and comprehensive sexual and reproductive health education. This is crucial for increasing self-confidence and enabling them to make informed decisions about their sexual and reproductive health. Furthermore, creating an enabling environment that respects and safeguards women's access to post-abortion care is warranted. This should also be expanded to the full extent of the country's legal framework and added as an integral component of existing sexual and reproductive health services.

## Supporting information

**S1 Table. The STROBE guidelines for cross-sectional studies.**
(DOCX)

## Acknowledgments

The authors would like to thank the DHS program and its partners for permitting them to analyze the dataset.

## Author Contributions

**Conceptualization:** McKenzie Maviso.

**Data curation:** McKenzie Maviso.

**Formal analysis:** McKenzie Maviso.

**Methodology:** McKenzie Maviso, Paula Zebedee Aines, Glen Mola, John W. Bolnga.

**Software:** McKenzie Maviso.

**Validation:** Paula Zebedee Aines, Gracelyn Potjepat, John W. Bolnga.

**Visualization:** McKenzie Maviso, Paula Zebedee Aines, Gracelyn Potjepat, John W. Bolnga.

**Writing – original draft:** McKenzie Maviso.

**Writing – review & editing:** McKenzie Maviso, Paula Zebedee Aines, Gracelyn Potjepat, Nancy Geregl, Glen Mola, John W. Bolnga.

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
