## [Decision Letter · Decision Letter 0]

12 Mar 2024

PONE-D-23-24906Prevalence of pregnancy termination and associated factors among married women in Papua New Guinea: a nationally representative cross-sectional surveyPLOS ONE

Dear Dr. Maviso,

Thank you for submitting your manuscript to PLOS ONE. After careful consideration, we feel that it has merit but does not fully meet PLOS ONE’s publication criteria as it currently stands. Therefore, we invite you to submit a revised version of the manuscript that addresses the points raised during the review process.

We look forward to receiving your revised manuscript.

Kind regards,

Blessing Akombi-Inyang, Ph.D.

Academic Editor

PLOS ONE

Reviewers' comments:

Reviewer's Responses to Questions

**Comments to the Author**

1. Is the manuscript technically sound, and do the data support the conclusions?

Reviewer #1: Yes

Reviewer #2: Yes

2. Has the statistical analysis been performed appropriately and rigorously? 

Reviewer #1: No

Reviewer #2: Yes

3. Have the authors made all data underlying the findings in their manuscript fully available?

Reviewer #1: Yes

Reviewer #2: No

4. Is the manuscript presented in an intelligible fashion and written in standard English?

Reviewer #1: No

Reviewer #2: No

5. Review Comments to the Author

Reviewer #1: The study discusses the prevalence and associated factors of pregnancy termination among married women aged 15–49 in Papua New Guinea (PNG), where abortion is not decriminalized and access to safe abortion services is scarce. The study concludes that pregnancy termination rates remain high in PNG, highlighting the urgent need for health interventions and support services to prevent unwanted pregnancies and unsafe abortion practices. Emphasizing the necessity of expanding access to safe abortion services and integrating them into existing sexual and reproductive healthcare is crucial for addressing this issue comprehensively. Although it is a very relevant and important report for the current issue surrounding unsafe abortion in PNG, there are some points that should be addressed before further evaluation for publication, here are some comments to improve the manuscript:

1- Line 41: Please add «married» women as the sample is restricted to those populations if it is the total number of women who participated in this survey, then you should mention it and also include the age group.

2- Line 147-149: you have mentioned that women who were married or in a formal union and responded to whether they had ever terminated a pregnancy with complete information on all factors of interest were eligible to participate. Since you have used secondary data and not conducted the survey, it is better to revise the sentence and write you have only included the married or in formal union women who had ever terminated a pregnancy with complete information on all factors of interest.

3- you have only included the married or in formal union women, I believe if traditionally and culturally, marriage is the only accepted practice to get pregnant in PNG then the risk of illegal abortion and complications would be higher than those who are in formal relation such as girl friend and boy friend or cohabiting partners that have not married. My recommendation would be to separate them and do a separate analysis for these two groups and see if there is a difference unless you have a reasonable explanation for why you have included both.

4- I strongly believe that you should exclude those who are less than 18 years old as they are considered children by definition and illegal abortion, risk factors, and complications are often more prevalent among them especially if they are exposed to violence and discrimination in informal relationships.

5- line 165: education and literacy are different concepts, therefore I advise including literacy. It has been shown that women with literacy even with no formal education have a better pregnancy outcome. Some of those with no education still have literacy of reading and writing

5- another important variable could be the husband or partner's age, education, and employment. I suggest to include it

6- how the missing data were handled?

7- the analysis strategy for multivariate analysis including the choice of the independent variables that included in the models and the technique that has been used to do so as well as the strategy to construct the final optimal model should be explained

8-

Reviewer #2: Thank you dear editor for inviting me to review this manuscript which provides interesting findings for policy makers in the respective country. I have provided the following comments for the author.

1.Prevalence of pregnancy termination and associated factors among married women in Papua New Guinea- the tittle is informative and clearly understandable however the focus of your study is on induced abortion particularly unsafe abortion so why did you used the general term ‘’pregnancy termination’’?

2. What do you think about the non-response rate which is very high in this study so how did you manage it?

3. On the conclusion part you said the prevalence of pregnancy termination among married women remains high in PNG. What was your reference to say it is high?

4. On the data source line number 149 you said that the extracted data on these women included information on their socio-demographic characteristics and the history of termination of pregnancy within the three years preceding the survey. You did not exclude those mothers without the history of pregnancy within the three years. I raised this issue because it has no importance of including them in this study since they were not pregnant within the three years.

5. The line umber 159 ….Women were asked whether they had ever terminated a pregnancy. The response was coded as “0” for “No” and “1” for “Yes” for pregnancy termination. For this study, the terms pregnancy termination and induced abortion are used interchangeably. This part should be written in the operational definition.

6. PLOS authors have the option to publish the peer review history of their article (what does this mean?). If published, this will include your full peer review and any attached files.

Reviewer #1: **Yes: **Omid Dadras

Reviewer #2: **Yes: **Yosef Haile Gebremariam

---

## [Author Response · Author response to Decision Letter 0]

2 Jun 2024

Reviewer #1:

The study discusses the prevalence and associated factors of pregnancy termination among married women aged 15–49 in Papua New Guinea (PNG), where abortion is not decriminalized and access to safe abortion services is scarce. The study concludes that pregnancy termination rates remain high in PNG, highlighting the urgent need for health interventions and support services to prevent unwanted pregnancies and unsafe abortion practices. Emphasizing the necessity of expanding access to safe abortion services and integrating them into existing sexual and reproductive healthcare is crucial for addressing this issue comprehensively. Although it is a very relevant and important report for the current issue surrounding unsafe abortion in PNG, there are some points that should be addressed before further evaluation for publication, here are some comments to improve the manuscript:

Response: Thank you for your time in reviewing our manuscript. Your comments and suggestions are very important and have been taken into consideration to improve the paper. We have looked through the areas and sections highlighted and made the necessary changes and corrections which are now reflected in the manuscript.

1- Line 41: Please add «married» women as the sample is restricted to those populations if it is the total number of women who participated in this survey, then you should mention it and also include the age group.

Response: Thank you for highlighting this oversight. We have now included “married” and is reflected in the text. 

2- Line 147-149: you have mentioned that women who were married or in a formal union and responded to whether they had ever terminated a pregnancy with complete information on all factors of interest were eligible to participate. Since you have used secondary data and not conducted the survey, it is better to revise the sentence and write you have only included the married or in formal union women who had ever terminated a pregnancy with complete information on all factors of interest.

Response: Thank you for highlighting this to improve the clarity of the sentence. The sentence has been revised. It is corrected and is reflected in the text.

3- you have only included the married or in formal union women, I believe if traditionally and culturally, marriage is the only accepted practice to get pregnant in PNG then the risk of illegal abortion and complications would be higher than those who are in formal relation such as girlfriend and boyfriend or cohabiting partners that have not married. My recommendation would be to separate them and do a separate analysis for these two groups and see if there is a difference unless you have a reasonable explanation for why you have included both.

Response: Thank you for the recommendation. Our analysis focused mainly on those women who reported that they were married and not in any boy-girl relationships or as cohabiting partners. Also, the demographic and health survey (DHS) data did not specifically ask about the type of relationships the women were in, during the time of the survey. The survey questions only asked whether they were married or not. Some women who were married and were divorced, separated, or widowed were recorded. Only those women who reported being married were included in this study. We, therefore could explore the determinants of pregnancy termination among women in informal (unmarried) relationships in future analyses, as presumptively, these group could yield higher results of pregnancy termination.

4- I strongly believe that you should exclude those who are less than 18 years old as they are considered children by definition and illegal abortion, risk factors, and complications are often more prevalent among them especially if they are exposed to violence and discrimination in informal relationships.

Response: Thank you for highlighting this for further explanation. It is established that marriage in most societies in PNG is closely linked to socio-cultural influences. In most instances, women often marry at an early age (>18 years). It is possible that women were unsure of their date of birth and indicated the wrong age during the survey, especially young married women from rural areas. The survey data lacked a specific age of marriage, and we are unsure whether women were married too early (<18 years); therefore, we only used the age-aggregated variable (V013).

5- line 165: education and literacy are different concepts; therefore, I advise including literacy. It has been shown that women with literacy even with no formal education have a better pregnancy outcome. Some of those with no education still have literacy of reading and writing

Response: Thank you for the suggestion. We have included the variable literacy.

5- another important variable could be the husband or partner's age, education, and employment. I suggest to include it.

Response: Thank you for the suggestion. We have included the variable partner’s age.

6- how the missing data were handled?

Response: Thank you for seeking clarification. Data were missing at random, which were not evident across all observations, but only within sub-samples of the data. Those missing data in variables of interest that have low percentage (<5%) were dropped, while those with high percentage (>5%) were retained for final analysis. This is reflected in the results (table 1 and Table 2).

7- the analysis strategy for multivariate analysis including the choice of the independent variables that included in the models and the technique that has been used to do so as well as the strategy to construct the final optimal model should be explained

Response: Thank you for the suggestion. The independent variables were selected based on their availability in the dataset, practical significance, and relevance reported in the literature about pregnancy termination. Since the study used data collected by a cluster sampling technique, a complex sample analysis was deemed a suitable technique. Also, complex sample analysis provides valid estimates of parameters because, during analysis, it accounts for sample weighting, clustering, and stratification. Thus, a complex sample logistic regression was used to perform a multivariable analysis of predictors of pregnancy termination. This is explained in the methods.

Reviewer #2:

Thank you dear editor for inviting me to review this manuscript which provides interesting findings for policy makers in the respective country. I have provided the following comments for the author.

Response: Thank you for your time in reviewing our manuscript. Your comments and suggestions are taken into consideration to improve our paper. We have looked through the areas and sections highlighted and made the necessary changes and corrections, and are all reflected in the manuscript.

1. Prevalence of pregnancy termination and associated factors among married women in Papua New Guinea- the tittle is informative and clearly understandable however the focus of your study is on induced abortion particularly unsafe abortion so why did you used the general term ‘’pregnancy termination’’?

Response: Thank you for seeking clarification. We have used the term “pregnancy termination” throughout this analysis, as this is probably the softer and more non-discriminatory way of describing the issue (euphemism).We have therefore included an operational definition as suggested.

2. What do you think about the non-response rate which is very high in this study so how did you manage it?

Response: Thank you for seeking clarification. The design weights were adjusted for individual non-response to get the sampling weights for women. Non-response is adjusted at the sampling stratum level. After adjusting for non-response, the sampling weights are normalized to get the final standard weights that appear in the data for analysis.

3. On the conclusion part you said the prevalence of pregnancy termination among married women remains high in PNG. What was your reference to say it is high?

Response: Thank you for highlighting this to seek further clarification. We acknowledge the statement as on oversight. The sentence has been revised and corrected. It is reflected in the conclusion: Pregnancy termination is performed among married women in PNG.

4. On the data source line number 149 you said that the extracted data on these women included information on their socio-demographic characteristics and the history of termination of pregnancy within the three years preceding the survey. You did not exclude those mothers without the history of pregnancy within the three years. I raised this issue because it has no importance of including them in this study since they were not pregnant within the three years.

Response: Thank you for highlighting this oversight. We have revised and dropped those married women who have no history of pregnancy in the past 3 years. It has been corrected and is reflected in the text.

5. The line umber 159 …. Women were asked whether they had ever terminated a pregnancy. The response was coded as “0” for “No” and “1” for “Yes” for pregnancy termination. For this study, the terms pregnancy termination and induced abortion are used interchangeably. This part should be written in the operational definition.

Response: Thank you. Yes, we have included it as operational definition for this study. It is reflected in the manuscript.

---

## [Decision Letter · Decision Letter 1]

21 Aug 2024

Prevalence of pregnancy termination and associated factors among married women in Papua New Guinea: a nationally representative cross-sectional survey

PONE-D-23-24906R1

Dear Dr. Maviso,

We’re pleased to inform you that your manuscript has been judged scientifically suitable for publication and will be formally accepted for publication once it meets all outstanding technical requirements.

Kind regards,

Amos Buh, BSc., MPH, PhD

Academic Editor

PLOS ONE

Additional Editor Comments (optional):

Reviewers' comments:

Reviewer's Responses to Questions

**Comments to the Author**

1. If the authors have adequately addressed your comments raised in a previous round of review and you feel that this manuscript is now acceptable for publication, you may indicate that here to bypass the “Comments to the Author” section, enter your conflict of interest statement in the “Confidential to Editor” section, and submit your "Accept" recommendation.

Reviewer #2: All comments have been addressed

2. Is the manuscript technically sound, and do the data support the conclusions?

Reviewer #2: Yes

3. Has the statistical analysis been performed appropriately and rigorously? 

Reviewer #2: Yes

4. Have the authors made all data underlying the findings in their manuscript fully available?

Reviewer #2: No

5. Is the manuscript presented in an intelligible fashion and written in standard English?

Reviewer #2: No

6. Review Comments to the Author

Reviewer #2: (No Response)

7. PLOS authors have the option to publish the peer review history of their article (what does this mean?). If published, this will include your full peer review and any attached files.

Reviewer #2: **Yes: **Yosef Haile Gebremariam

---

## [Editor Report · Acceptance letter]

26 Aug 2024

PONE-D-23-24906R1 

PLOS ONE

Dear Dr. Maviso, 

I'm pleased to inform you that your manuscript has been deemed suitable for publication in PLOS ONE. Congratulations! Your manuscript is now being handed over to our production team.

Kind regards, 

on behalf of

Dr. Amos Buh 

Academic Editor

PLOS ONE